# Targeting Mitochondrial Oxidative Stress as a Strategy to Treat Aging and Age-Related Diseases

**DOI:** 10.3390/antiox12040934

**Published:** 2023-04-15

**Authors:** Yun Haeng Lee, Myeong Uk Kuk, Moon Kyoung So, Eun Seon Song, Haneur Lee, Soon Kil Ahn, Hyung Wook Kwon, Joon Tae Park, Sang Chul Park

**Affiliations:** 1Division of Life Sciences, College of Life Sciences and Bioengineering, Incheon National University, Incheon 22012, Republic of Korea; 2Convergence Research Center for Insect Vectors, Incheon National University, Incheon 22012, Republic of Korea; 3The Future Life & Society Research Center, Chonnam National University, Gwangju 61186, Republic of Korea

**Keywords:** mitochondrial oxidative stress, mitochondria, ROS, aging control

## Abstract

Mitochondria are one of the organelles undergoing rapid alteration during the senescence process. Senescent cells show an increase in mitochondrial size, which is attributed to the accumulation of defective mitochondria, which causes mitochondrial oxidative stress. Defective mitochondria are also targets of mitochondrial oxidative stress, and the vicious cycle between defective mitochondria and mitochondrial oxidative stress contributes to the onset and development of aging and age-related diseases. Based on the findings, strategies to reduce mitochondrial oxidative stress have been suggested for the effective treatment of aging and age-related diseases. In this article, we discuss mitochondrial alterations and the consequent increase in mitochondrial oxidative stress. Then, the causal role of mitochondrial oxidative stress on aging is investigated by examining how aging and age-related diseases are exacerbated by induced stress. Furthermore, we assess the importance of targeting mitochondrial oxidative stress for the regulation of aging and suggest different therapeutic strategies to reduce mitochondrial oxidative stress. Therefore, this review will not only shed light on a new perspective on the role of mitochondrial oxidative stress in aging but also provide effective therapeutic strategies for the treatment of aging and age-related diseases through the regulation of mitochondrial oxidative stress.

## 1. Introduction

When somatic cells reach a certain number of mitotic levels, they begin to lose their ability to proliferate, which is one of the hallmarks of senescence [1]. Along with losing the capacity to divide, senescent cells show dramatic changes, such as enlarged and flattened cell morphology, increased production of reactive oxygen species (ROS), the accumulation of consequent ROS-mediated damage derivatives (e.g., lipofuscin and granules), and the senescence-associated secretory phenotype (SASP) [2]. Senescent cells accumulate with age, negatively impacting regenerative capacity and creating a proinflammatory environment conducive to the onset and development of aging and age-related diseases [2]. This causal relationship is supported by the finding that the selective removal of senescent cells in vivo reduces inflammation and improves immune system function, slowing the development of aging and thereby extending lifespan [3]. Furthermore, senolytic therapies that eliminate senescent cells prevent age-related bone loss and fragility, supporting the causal link between senescence and aging [4].

Changes in organelle morphology or function are another characteristic of senescence, among which mitochondrial degeneration is most prominent [5,6]. Mitochondria exhibit structural changes such as significant increases in volume and size due to the buildup of defective mitochondria [7]. Defective mitochondria generate ROS as a byproduct of electron leakage from the electron transport complex (ETC) [8]. Not only are defective mitochondria ROS generators, but they are also targets of mitochondrial oxidative stress, which then boosts mitochondrial ROS production. Mitochondrial ROS generated by defective mitochondria deteriorate the morphology and function of organelles, consequently leading to aging and age-related diseases [9]. Therefore, strategies to reduce mitochondrial oxidative stress may be beneficial as therapeutic approaches to aging and age-related diseases [10,11]. The finding that treatment of senescent cells with ROS scavengers restored the senescent phenotype supports the usefulness of this strategy [11]. Mitochondrial oxidative stress is a major cause of senescence and the consequent development of age-related diseases, so a deeper comprehension of the mechanisms that target and control mitochondrial oxidative stress is needed.

This review discusses mitochondrial alterations and the consequent increase in mitochondrial oxidative stress and proposes ways to reduce mitochondrial oxidative stress to treat aging and age-related diseases. Using search terms including “mitochondrial alterations”, “mitochondrial oxidative stress”, and “mitochondrial ROS”, a thorough literature search was conducted in PubMed (a database of life sciences and medical journal articles). Based on previous and current studies derived from the literature, we provide a new perspective on the causal link between mitochondrial oxidative stress and aging and suggest potential therapeutic options for treating aging and age-related diseases.

## 2. Mitochondrial Alterations during the Process of Senescence and Aging

Mitochondria maintain their morphology, quality, and function through mitochondrial dynamics consisting of fusion and fission (Figure 1A). Mitochondrial fusion occurs in two steps. Mitofusin 1 (Mfn1) and mitofusin 2 (Mfn2), mitochondrial dynamin-like GTPases, fuse the outer mitochondrial membrane (OMM) [12,13] (Figure 1A). Then, OPA1 mitochondrial dynamin-like GTPase fuses the inner mitochondrial membrane (IMM). Mitochondrial fusion allows the mixing of the components of healthy and partially damaged mitochondria, resulting in a mitochondrial network with more uniform components [14]. Mitochondrial fission is regulated by receptors on OMM, including a mitochondrial dynamics protein of 49 kDa and 51 kDa (MiD49 and MiD51, respectively), mitochondrial fission factor (Mff), and fission 1 protein (FIS1) [15] (Figure 1A). The endoplasmic reticulum (ER) initiates mitochondrial fission by constricting the mitochondrial membrane, and then MiD49 and MiD51 recruit dynamin-related protein 1 (Drp1) to the mitochondrial surface [16,17] (Figure 1A). This binding allows higher-order Drp1 oligomers to form around the mitochondrial surface, leading to separation into two mitochondria [15,17]. A recent study subdivides mitochondrial fission into fission occurring in the mid-zone and periphery [18]. Mitochondrial fission at the mid-zone occurs when ER tubules contact mitochondria and constrict [18]. Then, Mff recruits Drp1 to scission sites. Mitochondrial fission at the periphery precedes lysosomal contact and is regulated by FIS1, which recruits Drp1 to scission sites [18]. Both types of mitochondrial fission are mediated by Drp1, inducing the formation of higher-order Drp1 oligomers around the scission site, splitting one mitochondrion into two mitochondria [18]. Mitochondrial fission at the mid-zone creates new mitochondria to provide necessary mitochondria during cell growth and division [18]. Mitochondrial fission at the periphery serves as quality control by isolating defective mitochondria from the mitochondrial network and allowing them to be eliminated by mitochondria-specific autophagy (mitophagy) [18]. Senescent cells show changes in mitochondrial morphology, such as increased mitochondrial mass and size [19,20]. Specifically, age-related lysosomal dysfunction prevents lysosomes from fusing with autophagosomes, limiting the efficient clearance of defective mitochondria via mitophagy and leading to the accumulation of defective mitochondria with aberrant and large morphology [21,22]. The causal role of senescence on mitochondrial morphology is corroborated by the finding that both senescent and H_2_O_2_-induced senescent cells express low levels of FIS1, resulting in an imbalance in mitochondrial fusion–fission [23] (Figure 1B and Table 1). This imbalance promotes the formation of large mitochondria with highly interconnected network structures [23] (Figure 1B and Table 1). This observation is supported by the discovery that senescent cells exhibit lower FIS1 and Drp1 expression, mediating the formation of large mitochondria and the resistance to oxidative stress through PTEN-induced putative kinase protein 1 (PINK1) [24]. Furthermore, FIS1 knockdown cells exhibit large mitochondria with concomitant senescence-related phenotypic changes [25]. Large mitochondria also limit the efficiency of clearing damaged mitochondria through mitophagy, reducing mitochondrial turnover. This phenomenon is evidenced by the finding that a 60% increase in mitochondrial size and a considerable rise in the percentage of large mitochondria were observed in cells from aged mice [26].

Mitochondrial homeostasis is primarily regulated by mitochondrial Ca^2+^ concentration. Ca^2+^ enters the mitochondria from the ER through mitochondrial porins known as voltage-dependent anion channels (VDACs) in the OMM (Figure 2A). Specifically, G protein-coupled receptor 75 (Gpr75) connects the inositol 1,4,5-triphosphate receptor (IP_3_R) in the ER with VDACs, facilitating Ca^2+^ influx from the ER into the mitochondria [27,28,29] (Figure 2A). Ca^2+^ passes via the mitochondrial calcium uniporter (MCU) in the IMM and is then taken up into the mitochondrial matrix. On the other hand, Ca^2+^ efflux occurs through channels that are distinct from those of Ca^2+^ influx. The Na^+^/Ca^2+^ exchanger (NCLX) and H^+^/Ca^2+^ exchanger (HCX) found in the IMM are responsible for Ca^2+^ efflux (Figure 2A). Senescence is characterized by Ca^2+^-overloaded mitochondria and is exacerbated by mitochondrial Ca^2+^-overload-induced mitochondrial oxidative stress [30,31,32,33] (Figure 2B and Table 1). Specifically, during oncogene-induced senescence, IP_3_R triggers a sustained increase in IP_3_R-mediated Ca^2+^ release [31]. Ca^2+^ then starts to be transported to the mitochondria through the VDAC/MCU channel [31] (Figure 2B and Table 1). Mitochondrial Ca^2+^ overload induces mitochondrial ROS generation and senescence [31]; the detailed mechanism will be further discussed in Section 3.

**Table 1 antioxidants-12-00934-t001:** A summary of senescence-associated mitochondrial alterations.

Mitochondrial Alteration	Outcome(s)	Experimental Model and References
Mitochondrial morphology	Formation of large mitochondria with highly interconnected network structures	MRC-5 human embryonic lung fibroblasts [19,20]
	A considerable rise in the percentage of large mitochondria	C57/BL6 mice aged 30 months [26]
Mitochondrial Ca^2+^ homeostasis	Senescent cells show a sustained increase in IP_3_R-mediated Ca^2+^ release	Human endometrium-derived stem cells [30]
	Ca^2+^ then starts to be transported to the mitochondria through the VDAC/MCU channel	Human endometrial adenocarcinoma cells and WI38 human fibroblasts [31]
	Mitochondria overloaded with Ca^2+^ exhibit increased electron leakage from the ETC and consequently generate mitochondrial ROS	Human endometrial adenocarcinoma cells and WI38 human fibroblasts [31]

## 3. Defective Mitochondria Are a Major Cause of Mitochondrial ROS Generation 

More than 90% of oxygen is utilized by the mitochondria, and in complexes I and III of the ETC, 1–5% of oxygen is converted into short-lived superoxide anions (O_2_^−^) [34]. Complexes I and III convert oxygen to superoxide anions (O_2_^−^) in the mitochondrial matrix. In addition, complex III generates superoxide anions (O_2_^−^) in the mitochondrial intermembrane space. The aging-related deterioration of complex I activity occurs due to decreased expression and increased oxidation of complex I [35,36]. Consistent with these findings, neurodegenerative diseases such as Parkinson’s disease (PD) have reduced complex I activity, leading to increased mitochondrial ROS production and complex I oxidation [37,38,39,40]. Complex I damaged by oxidative stress inefficiently transports electrons and subsequently increases electron leakage to oxygen, generating superoxide anions (O_2_^−^) (Figure 3A and Table 2). Superoxide dismutase (SOD) converts superoxide anions (O_2_^−^) to hydrogen peroxide (H_2_O_2_), a non-radical derivative that is relatively stable and permeable to the mitochondrial membrane. Then, hydrogen peroxide (H_2_O_2_) is partially reduced by the Fenton reaction and converted to a more harmful free radical, the hydroxyl radical (^•^OH), which eventually causes severe mitochondrial oxidative stress [41] (Figure 3A). During the Fenton reaction, mitochondrial ROS is generated by mitochondrial iron that can be used for heme and iron–sulfur (Fe–S) cluster biosynthesis [42]. Mitochondrial dysfunction manifests as defects in heme and Fe–S cluster biosynthesis [43,44]. Alterations in iron homeostasis due to these defects lead to mitochondrial iron overload, resulting in the overproduction of free radicals via the Fenton reaction [45,46,47] (Table 2). 

Mitochondrial Ca^2+^ overload is one of the main causes of mitochondrial ROS production. Specifically, mitochondrial Ca^2+^ overload stimulates the mitochondrial permeability transition (PT) and opens PT pores (mPTP), allowing ions and other solutes to move freely [48,49] (Figure 3B). The osmotic pressure of the mitochondrial matrix is increased by mPTP opening, which causes swelling of the mitochondria. Enlarged mitochondria readily lose cytochrome c that is loosely bound to the IMM [50,51,52,53] (Figure 3B and Table 2). Because cytochrome c plays a key role in electron transport in the ETC, the loss of cytochrome c impedes electron transport from complex I to IV, resulting in increased electron leakage (Figure 3B and Table 2). The leaked electrons react with oxygen to generate large amounts of mitochondrial ROS [50,51]. The induced increase in mitochondrial ROS generation causes mPTP to open more frequently and for longer periods of time [54,55]. The sustained opening of mPTP triggers a rapid decrease in the mitochondrial membrane potential and ATP synthesis. Furthermore, the continuous influx of ions and other solutes through mPTP causes electron leakage, which increases mitochondrial ROS production [53,55,56] (Figure 3B). Mitochondrial Ca^2+^ overload also increases hydrogen peroxide (H_2_O_2_) formation in mitochondria through the activation of α-glycerophosphate dehydrogenase (α-GPDH) located on the outer surface of the IMM [32]. This discovery is corroborated by the observation that the activation of α-GPDH by mitochondrial Ca^2+^ increases mitochondrial ROS production by providing more electrons to the ETC and creating favorable conditions for reverse electron transport [33].

**Table 2 antioxidants-12-00934-t002:** Defective mitochondria are a major cause of mitochondrial ROS generation.

Cause of Mitochondrial ROS Generation	Outcome(s)	Experimental Model and References	ROS Related Information
Defective mitochondria	Neurodegenerative diseases such as PD have reduced complex I activity, leading to increased mitochondrial ROS production and complex I oxidation	Fibroblasts from the patient with PINK1 mutation [40]	ROS sources: mitochondrial ROS Type of ROS: superoxide anions (O_2_^−^), hydrogen peroxides (H_2_O_2_) Enzymes involved in ROS generation: complex I
	Mitochondrial dysfunction manifests as defects in heme and Fe–S cluster biosynthesis. Alterations in iron homeostasis due to these defects lead to mitochondrial iron overload, resulting in the overproduction of free radicals via the Fenton reaction	Chondrocyte C-20/A4 cell lines [47]	ROS sources: intracellular ROS Type of ROS: total ROS Enzymes involved in ROS generation: not specified
Mitochondrial Ca^2+^ overload	Mitochondrial Ca^2+^ overload stimulates the mitochondrial permeability transition (PT) and opens PT pores (mPTP), allowing ions and other solutes to move freely	Isolated heart mitochondria from the bovine heart [48,49]	
	The osmotic pressure of the mitochondrial matrix is increased by mPTP opening, which causes swelling of the mitochondria. Enlarged mitochondria readily lose cytochrome c that is loosely bound to the IMM	Astrocytes newborn C57BL/6 mice [53]	ROS sources: intracellular ROS Type of ROS: total ROS Enzymes involved in ROS generation: not specified
	Loss of cytochrome c impedes electron transport from complex I to IV, resulting in increased electron leakage from the ETC. The leaked electrons react with oxygen to generate large amounts of mitochondrial ROS	Isolated heart mitochondria from male Wistar rats [50]	ROS sources: mitochondrial ROS Type of ROS: hydrogen peroxides (H_2_O_2_) Enzymes involved in ROS generation: not specified

## 4. Vicious Feedback Loop between Mitochondrial Oxidative Stress and Senescence/Aging

ROS at physiologically low-to-moderate concentrations play biological roles in differentiation and proliferative responses [57,58]. For example, mild doses of oxidative stress induced by dietary restriction have been demonstrated to extend lifespan in various model organisms [59,60]. These results suggest that mild doses of oxidative stress can reduce chronic oxidative damage by increasing endogenous antioxidant defenses, triggering an adaptive response that enhances overall stress resistance [61]. Not only the level of ROS is important for cellular function, but the place where ROS is generated is also important. Mitochondrial ROS generation in complex I via reverse electron transport serves as a catalyst for preserving mitochondrial function and lengthening the lifespan in *Drosophila* [62]. However, in the same study, the mitochondrial ROS generation by preventing the activity of coenzyme Q (CoQ), which transports electrons from complex II to III, accelerates aging and shortens the lifespan in *Drosophila* [62]. These two conflicting results suggest that further studies are needed to understand how each ROS-producing site determines lifespan in *Drosophila* and whether each site plays a similar role in other organisms.

The physiological levels of ROS mediate important cellular functions, as shown in several studies, whereas the pathological levels of ROS cause irreversible damage to DNA, RNA, lipids, and proteins, which is a major contributor to aging and age-related diseases [63]. For example, accumulating oxidative damage over time activates the p53 and retinoblastoma protein pathways, resulting in persistent cell cycle arrest and senescence [9,64,65,66,67] (Figure 4). These findings are strengthened by recent studies showing that mitochondrial ROS activates a nicotinamide adenine dinucleotide (NAD^+^)-consuming enzyme called polyADP-ribose polymerase 1 [68,69]. Increased NAD^+^ consumption by polyADP-ribose polymerase 1 significantly reduces NAD^+^ levels [68]. As NAD^+^ is a crucial regulator allowing cells to respond to environmental changes, such as genotoxic factors and oxidative stress [70], a decrease in NAD^+^ levels leads to cellular dysfunction and aggravates age-related pathologies [71,72,73] (Figure 4). The role of mitochondrial oxidative stress on aging is further substantiated by the finding in the Caenorhabditis elegans (*C. elegans*) aging model caused by a *mev-1* (complex II ortholog) mutation. The biochemical pathology of *mev-1* mutants includes two-fold more mitochondrial ROS production than the wild type [74]. Increased mitochondrial ROS levels in *mev-1* mutants led to premature age-dependent physiological alterations, including the accumulation of lipofuscin and protein carbonyl derivatives [75,76]. Moreover, the average and maximum lifespans of *mev-1* mutants in oxygenated conditions were significantly decreased compared with those of the wild type [74]. Support for this phenomenon is seen in other studies targeting complex II. An iron chelator, deferoxamine, decreases complex II activity by inhibiting the conversion of Fe–S clusters in complex II [77]. Decreased complex II activity increases mitochondrial ROS production, resulting in senescence-like growth arrest [77,78]. Similar to these findings, transforming growth factor β-1 (*TGF-β-1*) activates mitochondrial ROS production by inactivating complex IV activity. The accumulation of mitochondrial oxidative stress through *TGF-β-1*-mediated inhibition directly causes degenerative changes [79,80,81]. The effect of mitochondrial oxidative stress on aging is further supported by findings in mice lacking superoxide dismutase 1 (SOD1), a protein found in both the mitochondrial matrix and the mitochondrial intermembrane space [82]. The absence of SOD1 increases superoxide anion (O_2_^−^) production and consequent oxidative damage, resulting in symptoms commonly seen in aged mice, such as the premature loss of skeletal muscle mass [82].

Mitochondrial oxidative stress easily damages mitochondrial proteins involved in proteostasis, which is important for preserving and regulating protein quality in mitochondria [83]. Mitochondrial proteostasis consists of proteases that clearly damage mitochondrial proteins and chaperones that promote protein folding [84] (Figure 4). It has evolved to combat various mitochondrial stresses, including mitochondrial ROS-mediated damage. The impact of mitochondrial oxidative stress on mitochondrial proteostasis is shown in a PD mouse model [85]. The increase in ROS levels in the PD mouse model inactivates the Lon Peptidase 1, a mitochondrial protease, initiating an early event in PD pathogenesis [85]. The detrimental effects of mitochondrial oxidative stress on mitochondrial proteostasis are also shown in Alzheimer’s disease (AD) patients and transgenic AD mice [86]. Increased oxidative stress underlies the reduced activity of mitochondrial proteases, which aggravates AD pathogenesis by accumulating β-amyloid peptides in the mitochondria [86]. Furthermore, the prolonged disturbance of mitochondrial proteostasis by intracellular and mitochondrial ROS causes the accumulation of misfolded or aggregated mitochondrial proteins and triggers cellular senescence [87,88] (Figure 4).

Mitochondrial oxidative stress contributes to inflammation and proinflammatory secretory phenotypes (Figure 4). Specifically, superoxide anion (O_2_^−^) reacts with SOD in the mitochondrial matrix to produce H_2_O_2_, which can pass through the OMM and react with cytoplasmic targets. This response activates the secretion of inflammatory cytokines (TNF-α and IL-1β) and proinflammatory SASP (IL-6 and IL-8) [89,90,91]. Mitochondrial ROS-mediated inflammatory responses exacerbate senescence, promoting chronic inflammation and age-related diseases in the long term [92]. Support for the causal link is evident in the discovery that mitochondria-targeted depletion inhibits the mitochondrial ROS production and secretion of key SASP factors such as IL-6 and IL-8 [93].

Mitochondrial DNA (mtDNA) is a circular chromosome found inside the mitochondria. mtDNA contains 37 genes encoding 2 rRNAs, 22 tRNAs, and 13 mitochondrial proteins, all essential for mitochondrial homeostasis [94]. Mitochondrial oxidative stress directly damages mtDNA, as mitochondrial ROS is produced in the mtDNA-containing mitochondrial matrix [95] (Figure 4). The increased production of mitochondrial ROS correlates linearly with the accumulation of mtDNA mutations [96]. Furthermore, mutations in mtDNA reduce the expression of essential proteins required for the ETC, leading to the amplification of mitochondrial oxidative stress [97]. This amplification results in a vicious cycle between mitochondrial oxidative stress and organelle degeneration [97]. The causal relationship between mtDNA mutation and aging is substantiated by the finding that the accumulation of mtDNA damage reduces ATP levels and triggers the recycling of β-amyloid toxicity, ultimately exacerbating neurodegeneration [98,99,100]. Support for this causal connection is evident in the finding that oxidative damage to mtDNA, but not to nuclear DNA, is inversely associated with maximal longevity in the heart and brain of mammals [101].

## 5. Targeting Mitochondrial Oxidative Stress as a Therapeutic Strategy for Aging and Age-Related Diseases 

As described in Section 3 and Section 4, defective mitochondria are major sources of mitochondrial oxidative stress, and mitochondrial oxidative stress causes premature deterioration of tissue and organ function. The causal relationship suggests that the proper control of mitochondrial oxidative stress can be one of the effective therapeutic strategies for aging and age-related diseases. Here, we propose several therapeutic strategies targeting mitochondrial oxidative stress (Table 3).

Reducing mitochondrial oxidative stress with antioxidants represents an effective treatment strategy for aging. In mammals, an age-related decline in CoQ has been observed in the heart, liver, kidney, and skeletal muscle [102,103,104] (Figure 5A; green CoQ indicates CoQ deficiency). Cells deficient in CoQ exhibit electron leakage as electron transport in the ETC is hindered. Premature electron leakage from the ETC combines with oxygen to generate mitochondrial ROS [105] (Figure 5A). Considering that CoQ depletion triggers mitochondrial ROS production, mitochondria-targeted CoQ (MitoQ) treatment is tested in rat models of neurodegeneration [106]. As a mitochondria-specific antioxidant, MitoQ significantly lowers mitochondrial ROS production. Furthermore, MitoQ treatment is beneficial in treating neurodegeneration, showing reductions in mitochondrial swelling, cristae loss, and oxidative cell death [106] (Figure 5A; pink CoQ indicates high levels of CoQ). The benefit of CoQ supplementation in lowering mitochondrial oxidative stress is further supported by findings from an accelerated aging model using mice [107]. CoQ supplementation reduces mitochondrial ROS production by promoting the activity of complexes I and IV. Furthermore, a reduction in mitochondrial oxidative stress by CoQ supplementation delays the onset of age-related symptoms, indicating that therapeutics aimed at reducing mitochondrial oxidative stress might be effective [107]. In line with these findings, the treatment of senescent fibroblasts with N-acetylcysteine (NAC), a ROS scavenger, ameliorates senescence-associated phenotypes [11]. Moreover, antioxidant therapy with NAC ameliorates chondrocyte senescence and alleviates osteoarthritic phenotypes [108]. The justification of therapeutic strategies targeting mitochondrial oxidative stress is reinforced by experimental findings using SS-31, a cell-permeable antioxidant targeting the IMM [109]. Treatment with SS-31 boosts electron transport in the ETC, reducing the generation of hydroxyl and peroxynitrite free radicals in the mitochondria [109]. The reduction in mitochondrial oxidative stress by SS-31 has been implicated in delaying or reversing senescence in age-related diseases including AD and PD [110,111,112]. Additional experimental evidence for reducing mitochondrial oxidative stress with antioxidants is shown in findings using mice overexpressing mitochondria-targeted catalase (mCAT), an enzyme that breaks down hydrogen peroxide (H_2_O_2_) into H_2_O and O_2_ [113]. Two separate lines of mice overexpressing mCAT reduce hydrogen peroxide (H_2_O_2_) production and subsequent oxidative damage [114]. Moreover, these reductions delay cardiac pathology and cataract development, concomitant with a significant increase in the median and maximum lifespan [114].

The regulation of mitochondrial oxidative stress through the induction of mitophagy represents a promising and powerful strategy. Mitophagy, the selective degradation of damaged mitochondria through autophagy, is an essential mechanism to maintain mitochondrial homeostasis [115]. Specifically, the mitochondria damaged by high levels of mitochondrial ROS recruit PTEN-induced putative kinase protein 1 (PINK1) and ubiquitin ligase PARKIN, initiating the early phase of PINK1/PARKIN-dependent mitophagy [116] (Figure 5B). PINK1 activates PARKIN recruitment to the OMM for the ubiquitylation of OMM proteins in damaged mitochondria. The creation of polyubiquitin chains in OMM proteins recruits autophagy receptor proteins that interact with LC3 (microtubule-associated protein 1A/1B light chain 3) in the phagophore. Then, damaged mitochondria are eliminated by the subsequent formation of phagophores, autophagosomes, and autolysosomes, consequently resulting in reduced mitochondrial ROS production (Figure 5B). The significance of reducing mitochondrial oxidative stress by the activation of mitophagy has been validated by other findings. For example, the overexpression of PARKIN reduces mitochondrial oxidative stress with the protection of age-related loss of skeletal muscle in aged mice [117], whereas knocking out PARKIN accumulates markers of oxidative stress and deteriorates the contractile function of skeletal muscles [118]. Furthermore, PARKIN-mediated mitochondrial clearance suppresses mitochondrial ROS production and abrogates the development of senescence phenotypes such as pro-oxidant/inflammatory signaling and cyclin-dependent kinase inhibitors (p21 and p16) [93]. It is well known that PINK1 plays a key role in the interaction between OMM proteins and LC3 in the autophagosome, but recent studies have also shown that mitophagy can occur through a PINK1-independent mechanism. BNIP3 (BCL2 and adenovirus E1B 19-kDa-interacting protein 3) and NIP3-like protein X (NIX), which are localized to the OMM, serve as LC3 receptors, which contribute to the induction of mitophagy [119]. BNIP3 or NIX-mediated mitophagy also plays a crucial role in slowing and mitigating aging. For example, in *Drosophila*, the neuronal activation of BNIP3-mediated mitophagy delays systemic aging [120]. Moreover, NIX overexpression mitigates senescence by activating the degradation of mitochondrial proteins [121].

Consistent with these findings, mice that were given the mitophagy inducer trehalose were found to have reduced superoxide production and improved age-related atherosclerosis [122]. Similarly, therapy with the autophagy inducer lithium in *C. elegans* activates mitochondrial turnover with a decrease in mitochondrial oxidative stress [123]. A reduction in mitochondrial oxidative stress through lithium treatment consequently leads to an increase in the lifespan and healthspan without significantly changing death rates [123]. Furthermore, trehalose, a mitophagy inducer, has the effect of slowing down the aging process when it is administered to mice with age-related neurological symptoms induced by *Atg7* knockdown [124]. The importance of mitophagy induction for regulating mitochondrial oxidative stress is further emphasized by findings related to nicotinamide riboside (NR) supplementation. NR supplementation raises NAD^+^ levels, which are compromised in aged animals [71,72,73]. Elevated NAD^+^ levels activate sirtuin 1, an NAD^+^-dependent histone/protein deacetylase that uses NAD^+^ to deacetylate forkhead box O3 and peroxisome proliferator-activated receptor co-activator 1-α (*PGC-1α*), transcriptional regulators that activate the genes involved in mitophagy [125,126]. Activated mitophagy then removes damaged mitochondria and reduces mitochondrial ROS levels, consequently alleviating disease pathology and prolonging lifespan in animal models [125,126]. Recent studies have provided new evidence that mitophagy induction through the modulation of ataxia telangiectasia mutagenesis (ATM) activity might be an effective treatment option for aging [127,128]. The inhibition of ATM activity increases V_1_–V_0_ assembly in the V-ATPase proton pump found at the lysosomal membrane, allowing re-acidification in lysosomes [127]. Acidified lysosomes activate mitophagy to clear damaged mitochondria and consequently lower mitochondrial ROS levels [127]. Reducing mitochondrial oxidative stress by simultaneously regulating ATM activity restores various senescence-associated phenotypes to the level of young cells [127,128].

Reducing mitochondrial oxidative stress by directly targeting genes that regulate the activity of ETC components could be an alternative therapeutic strategy. Rho-associated protein kinase (ROCK) controls mitochondrial ROS production by regulating the connections between Rac1b and cytochrome c [129] (Figure 5C). Specifically, ROCK activation phosphorylates Rac1b, making it easier for Rac1b to interact with cytochrome c. This interaction steals electrons from cytochrome c and causes a partial reduction in oxygen, triggering mitochondrial ROS production [129]. The inhibition of ROCK activity prevents Rac1b from intercepting electrons from cytochrome c, enabling efficient electron transport from complex III to IV [129] (Figure 5C). Then, efficient electron transport prompts complex IV activity, thereby reducing mitochondrial ROS production (Figure 5C). Reducing mitochondrial oxidative damage by regulating ROCK activity improves poor growth and the early beginning of senescent phenotypes [129,130]. The significance of regulating the activity of ETC components to reduce mitochondrial stress is also supported by other studies. Long-term caloric restriction (CR) upregulates the activity of complex IV, which partially offsets electron leak and reduces mitochondrial ROS production [131]. CR extends average and maximum lifespan by delaying aging and preventing the development of age-related symptoms [132,133]. Furthermore, epigallocatechin 3-gallate (EGCG), a CR mimic, also mitigates mitochondrial oxidative stress and restores the catalytic activity of complex I/ATP synthase [134]. EGCG treatment ameliorates the dysfunction of age-related immune disorders and extends the lifespan in various animal models [135,136,137,138].

Maintaining mitochondrial ROS levels through mitochondrial Ca^2+^ homeostasis also represents an effective therapeutic option. As described in Section 2, adequate levels of mitochondrial Ca^2+^ maintain mitochondrial homeostasis [139], whereas mitochondrial Ca^2+^ overload generates more mitochondrial ROS [140]. Since fine-tuning the mitochondrial Ca^2+^ concentrations can maintain mitochondrial ROS at an appropriate level [141], various strategies have been attempted to control mitochondrial Ca^2+^. For instance, myocardial reperfusion is an age-related disease manifested by reduced resistance to cardiac reperfusion damage [142]. The major cause of myocardial reperfusion is cardiac tissue damage due to the increased mitochondrial ROS production resulting from increased mitochondrial Ca^2+^ levels in cardiomyocytes [143]. Since Ca^2+^ transport from the cellular cytoplasm to the mitochondrial matrix is promoted by MCU in the IMM [144,145], ruthenium 360 (Ru360), a cell-permeable MCU inhibitor, was administered to the mitochondria in the reperfused heart. The inhibition of MCU by Ru360 reduces the percentage of mitochondria exhibiting Ca^2+^ overload and subsequently reduces the production of mitochondrial ROS [146] (Figure 5D). Subsequently, the pathological signs of mitochondrial swelling return to a steady state with reduced mitochondrial ROS production [146]. The significance of minimizing mitochondrial oxidative damage by maintaining appropriate levels of mitochondrial Ca^2+^ is further evidenced by the finding that the microRNA-mediated silencing of MCU shields cardiomyocytes from mitochondrial oxidative stress [147] (Figure 5D).

Reducing mitochondrial oxidative stress through sirtuin modulation can be an alternative therapy to control aging. Sirtuins respond to mitochondrial oxidative stress through deacetylating transcription factors that control antioxidant genes. In particular, sirtuin 1 deacetylates and activates *PGC-1α*, which increases the function of genes that can regulate oxidative stress, such as catalase, glutathione peroxidase, and manganese SOD [148]. The reduction in mitochondrial ROS levels by sirtuin 1 ameliorates the symptoms of age-related neurodegeneration by protecting cultured neurons from oxidative stress-mediated death [148]. These findings are strengthened by other studies showing that therapy with resveratrol, a sirtuin 1 activator, reduces H_2_O_2_-induced mitochondrial oxidative damage and protects against H_2_O_2_-induced cell death [149]. Furthermore, the overexpression of sirtuin 2 extends the shortened lifespan caused by hydrogen peroxide (H_2_O_2_) treatment, supporting the claim that reducing oxidative stress by regulating sirtuins is an effective strategy to control aging [150].

**Table 3 antioxidants-12-00934-t003:** Targeting mitochondrial oxidative stress as a therapeutic strategy for aging and age-related diseases.

Therapeutic Strategy	Outcome(s)	Experimental Model and References
Antioxidants	Mitochondria-targeted CoQ (MitoQ) treatment lowers mitochondrial ROS production, consequently resulting in reductions in mitochondrial swelling, cristae loss, and oxidative cell death	Male albino rats (Wistar strain) [106]
	Treatment of senescent fibroblasts with N-acetylcysteine (NAC), a ROS scavenger, ameliorates senescence-associated phenotypes	Laminopathy progeria fibroblasts [11]
	Treatment with SS-31 boosts electron transport in the ETC, reducing the generation of free radicals in mitochondria	Primary neurons from C57BL/6 mice [109]
Mitophagy	Overexpression of PARKIN reduces mitochondrial oxidative stress with the protection of age-related loss of skeletal muscle in aged mice	Skeletal muscle of aged mice [117]
	Knocking out PARKIN accumulates markers of oxidative stress and deteriorates the contractile function of skeletal muscles	PARKIN knockout mice [118]
	Mice given the mitophagy inducer trehalose showed reduced superoxide production and improved age-related atherosclerosis	C57/BL6 mice aged 27–28 months [122]
	Therapy with the autophagy inducer lithium in *C. elegans* activates mitochondrial turnover with a decrease in mitochondrial oxidative stress	*C. elegans* [123]
	Inhibition of ATM activity activates mitophagy and consequently lowers mitochondrial ROS levels	Human fibroblasts [127]
Genes that regulate the activity of ETC components	Inhibition of the ROCK activity facilitates complex IV activity, thereby reducing mitochondrial ROS production	Progeria skin fibroblasts [129]
	Long-term caloric restriction (CR) upregulates the activity of complex IV, which partially offsets electron leak and reduces mitochondrial ROS production	Female Swiss Albino balb/c mice [131]
Mitochondrial Ca^2+^ homeostasis	Inhibition of MCU by Ru360 reduces the percentage of mitochondria exhibiting Ca^2+^ overload and subsequently reduces the production of mitochondrial ROS	Isolated brain mitochondria from male Wistar rats [146]
	MicroRNA-mediated silencing of MCU shields cardiomyocytes from mitochondrial oxidative stress	Rat cardiac myoblast H9c2 cell line [147]
Sirtuin	Reduction in mitochondrial ROS levels by sirtuin 1 ameliorates symptoms of age-related neurodegeneration by protecting cultured neurons from oxidative stress-mediated death	*PGC-1α* null cell [148]
	Overexpression of sirtuin 2 extends the shortened lifespan caused by hydrogen peroxide (H_2_O_2_) treatment	Yeast cells [150]

## 6. Conclusions and Perspectives

Targeting mitochondrial oxidative stress as a therapeutic strategy to treat aging and age-related diseases has been investigated and discussed in several review articles. One review article discussed the impacts of oxidative stress on mitochondrial function and aging [151]. However, the discussion was limited to a specific cell type, liver cells. Another review article did not limit the discussion to a specific type of cell but extended it to various models of aging [152]. Although the origins and effects of mitochondrial oxidative stress have been thoroughly investigated in various aging models, specific methods to reduce mitochondrial oxidative stress have not been proposed. A recent review article highlighted mitochondrial oxidative stress as a major cause of aging and a crucial determinant of longevity [153]. In that article, the methods to reduce mitochondrial ROS were discussed, but the strategy was exclusively focused on mitochondrial antioxidants. Given the fact that mitochondrial oxidative stress is induced through multiple cellular signaling pathways, it is unclear whether therapeutic strategies focused only on antioxidants will be beneficial in treating aging and age-related disorders.

In this review, we investigated and discussed mitochondrial alterations and the consequent increase in mitochondrial oxidative stress. In addition, by examining the process through which mitochondrial oxidative stress progresses aging and aging-related diseases, we found that mitochondrial oxidative stress acts as a vicious feedback loop for aging. Here, we suggested mitochondrial oxidative stress as a potential target for aging and presented several therapeutic options aimed at reducing mitochondrial oxidative stress. Mitochondrial alterations increase with age and consequently induce mitochondrial oxidative stress, which has been effectively controlled using genetic and pharmacological approaches. Therefore, the optimal regulation of mitochondrial oxidative stress that does not depend on a single treatment strategy will effectively treat the onset and development of aging in which multiple signaling pathways are impaired.

As explored in this review, therapeutic approaches to reduce mitochondrial oxidative stress have proven to be an important factor in treating aging and age-related diseases. However, clinical trials using non-mitochondria-targeted antioxidants have shown that non-mitochondria-targeted antioxidant therapies are not effective in the treatment of aging and age-related diseases [154,155]. To complement these clinical findings, mitochondria-targeting antioxidants have recently been applied to various animal models, and there is growing evidence that mitochondria-targeting antioxidants have beneficial effects on aging and age-related diseases [156,157]. Therefore, research on how to effectively deliver genetic and pharmacological therapeutics targeting mitochondria will provide good therapeutic clues to break the vicious cycle leading to aging and age-related diseases.

## Figures and Tables

**Figure 1 antioxidants-12-00934-f001:**
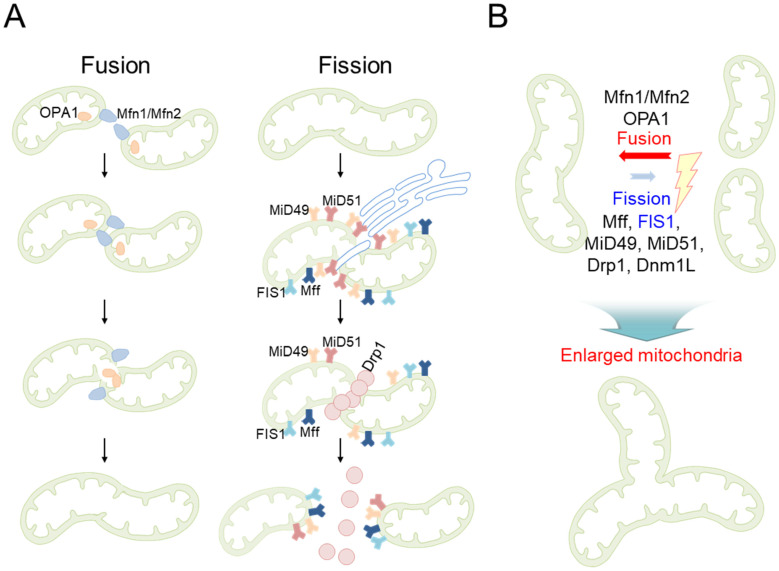
Mitochondrial alterations during the process of senescence and aging: (**A**) Mitochondria maintain their morphology, quality, and function through mitochondrial dynamics consisting of fusion and fission. Proteins involved in mitochondrial fusion: mitofusin 1 (Mfn1), mitofusin 2 (Mfn2), and OPA1. Proteins involved in mitochondrial fission: mitochondrial dynamics protein of 49 kDa and 51 kDa (MiD49 and MiD51, respectively), mitochondrial fission factor (Mff), mitochondrial fission 1 protein (FIS1), and dynamin-related protein 1 (Drp1). (**B**) Senescent and H_2_O_2_-induced senescent cells express low levels of FIS1, forming large mitochondria with highly interconnected network structures. ROS: reactive oxygen species. The lightning bolt represents senescence-associated stress.

**Figure 2 antioxidants-12-00934-f002:**
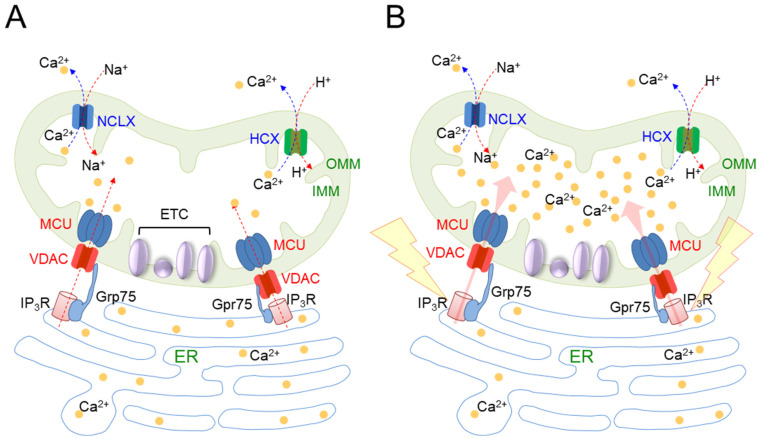
Fundamental mechanism of mitochondrial Ca^2+^ homeostasis: (**A**) Mitochondrial homeostasis is primarily regulated by mitochondrial Ca^2+^ concentration. Mitochondrial Ca^2+^ concentration is controlled by channels in the mitochondria and in the endoplasmic reticulum (ER). VDAC: voltage-dependent anion channels, MCU: mitochondrial calcium uniporter, HCX: H^+^/Ca^2+^ exchanger, NCLX: Na^+^/Ca^2+^ exchanger, IP_3_R: inositol 1,4,5-trisphosphate receptor. Orange dots represent Ca^2+^. (**B**) Senescence causes efflux of Ca^2+^ from the IP_3_R. Ca^2+^ then starts to be transported to the mitochondria through the VDAC/MCU channel. Mitochondrial Ca^2+^ overload induces mitochondrial ROS generation and senescence; the detailed mechanism will be further discussed in Section 3. Orange dots represent Ca^2+^. Lightning bolts represent senescence-associated stress. Pink arrows represent that large amounts of Ca^2+^ are being transported to the mitochondria through the VDAC/MCU channel.

**Figure 3 antioxidants-12-00934-f003:**
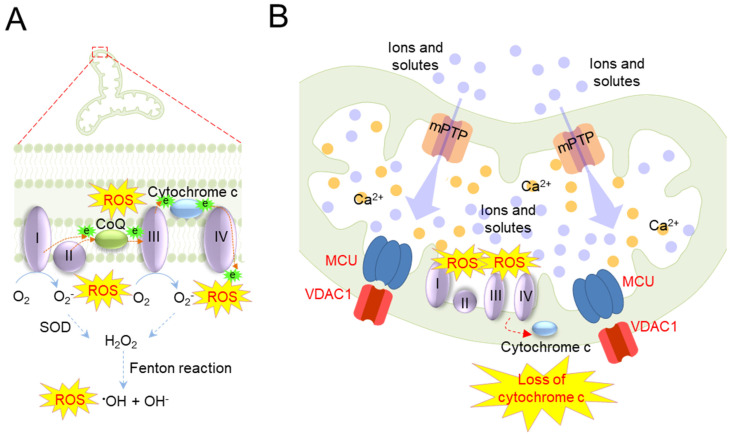
Defective mitochondria are a major cause of mitochondrial ROS generation: (**A**) Complex I damaged by oxidative stress inefficiently transports electrons and subsequently increases electron leakage to oxygen, generating superoxide anions (O_2_^−^). Superoxide dismutase (SOD) converts superoxide anions (O_2_^−^) to hydrogen peroxide (H_2_O_2_). Then, hydrogen peroxide (H_2_O_2_) is partially reduced by the Fenton reaction and converted to a more harmful free radical, the hydroxyl radical (^•^OH). e: electron. (**B**) Mitochondrial Ca^2+^ overload stimulates the mitochondrial permeability transition (PT) and opens PT pores (mPTP), allowing ions and other solutes to move freely. Enlarged mitochondria readily lose cytochrome c that is loosely bound to the IMM. Loss of cytochrome c impedes electron transport from complex I to IV, resulting in increased electron leakage from ETC. The leaked electrons react with oxygen to generate large amounts of mitochondrial ROS. Purple dots represent ions and solutes. Purple arrows represent that large amounts of ions and solutes are being transported to mitochondria via mPTP.

**Figure 4 antioxidants-12-00934-f004:**
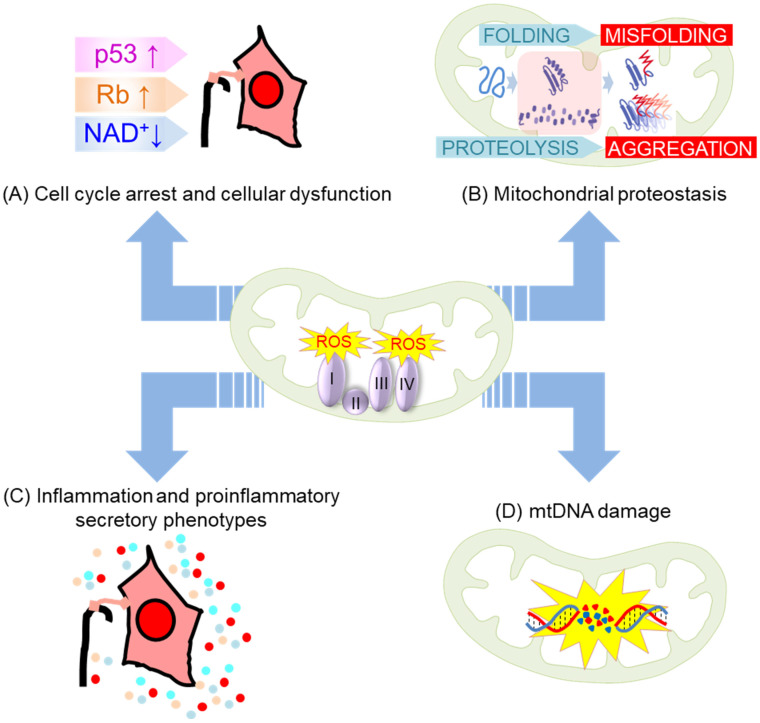
Vicious feedback loop between mitochondrial oxidative stress and senescence/aging. (**A**) Accumulating oxidative damage over time activates the p53 and Rb (retinoblastoma protein) pathways, resulting in persistent cell cycle arrest and senescence. Mitochondrial ROS also activates polyADP-ribose polymerase 1, which increases NAD^+^ consumption, thereby reducing NAD^+^ levels. A decrease in NAD^+^ levels leads to cellular dysfunction and aggravates age-related pathologies. (**B**) Mitochondrial proteostasis consists of chaperones that promote protein folding and proteases that clearly damage mitochondrial proteins. The prolonged disturbance of mitochondrial proteostasis by intracellular and mitochondrial ROS causes the accumulation of misfolded or aggregated mitochondrial proteins and triggers cellular senescence. (**C**) Mitochondrial oxidative stress activates the secretion of inflammatory cytokines and proinflammatory senescence-associated secretory phenotype (SASP). Different colored dots represent inflammatory cytokines and proinflammatory SASP. (**D**) Mitochondrial oxidative stress directly damages mtDNA, as mitochondrial ROS is produced in the mtDNA-containing mitochondrial matrix.

**Figure 5 antioxidants-12-00934-f005:**
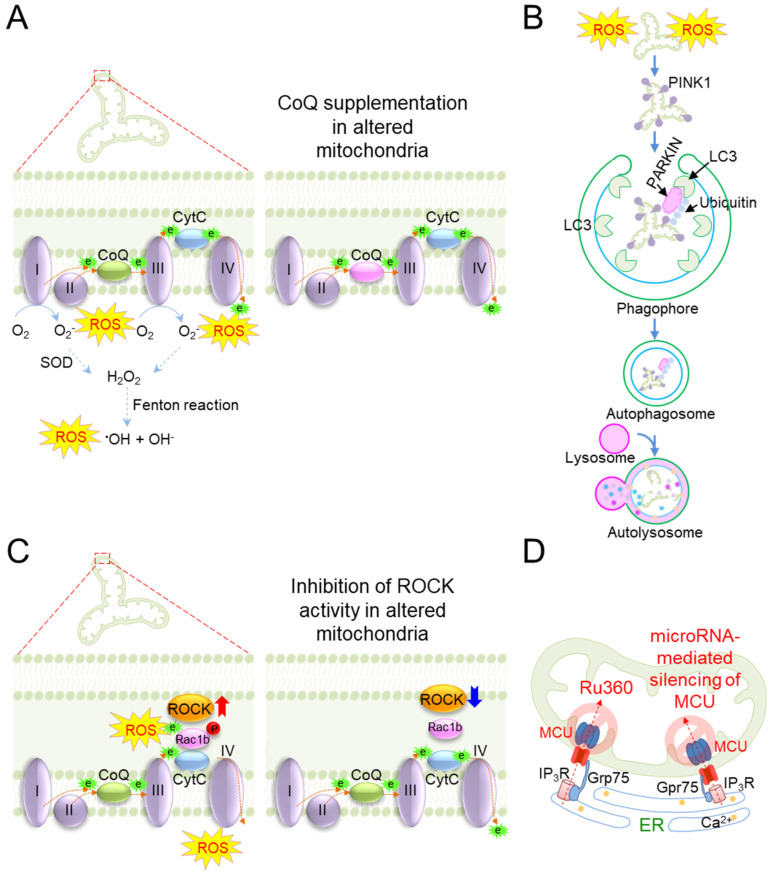
Targeting mitochondrial oxidative stress as a therapeutic strategy for aging and age-related diseases: (**A**) Senescent cells show a lack of coenzyme Q (CoQ), which transports electrons from complex II to III (green CoQ indicates CoQ deficiency). Senescent cells deficient in CoQ exhibit electron leakage as electron transport in ETC is hindered. Premature electron leakage from ETC combines with oxygen to generate mitochondrial ROS. By contrast, mitochondria-targeted CoQ (MitoQ) treatment lowers mitochondrial ROS production, consequently resulting in a significant reduction in senescence-associated symptoms (pink CoQ indicates high levels of CoQ); e: electron. (**B**) Mitochondria damaged by high levels of mitochondrial ROS recruit PTEN-induced putative kinase protein 1 (PINK1) and ubiquitin ligase PARKIN. PINK1 activates PARKIN recruitment to the OMM for ubiquitylation of OMM proteins in damaged mitochondria. The creation of polyubiquitin chains in OMM proteins recruits autophagy receptor proteins that interact with LC3 (microtubule-associated protein 1A/1B light chain 3) in the phagophore. Then, damaged mitochondria are eliminated by the subsequent formation of the phagophores, autophagosomes, and autolysosomes. (**C**) Reduction in mitochondrial oxidative damage by directly targeting genes that regulate the activity of ETC components. ROCK activation phosphorylates Rac1b, making it easier for Rac1b to interact with cytochrome c. This interaction steals electrons from cytochrome c and causes a partial reduction in oxygen, triggering mitochondrial ROS production. Inhibition of ROCK activity prevents Rac1b from intercepting electrons from cytochrome c, enabling electron transport from complex III to IV. Then, efficient electron transport facilitates complex IV activity, thereby reducing mitochondrial ROS production; e: electron. (**D**) Inhibition of MCU by ruthenium 360 (Ru360) reduces the percentage of mitochondria exhibiting Ca^2+^ overload and subsequently reduces the production of mitochondrial ROS. MicroRNA-mediated silencing of MCU shields cardiomyocytes from mitochondrial oxidative stress.

## Data Availability

Not applicable.

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
