# Peer review of "Targeting Mitochondrial Oxidative Stress as a Strategy to Treat Aging and Age-Related Diseases"

_antioxidants, 2023, doi:10.3390/antiox12040934_

Round 1
Reviewer 1 Report
This review discusses senescence-associated mitochondrial alterations and the resulting increase in mitochondrial oxidative stress. It discusses the causal role of mitochondrial oxidative stress on senescence, is investigating how senescence is exacerbated by induced stress, is assessing the importance of targeting mitochondrial oxidative stress for the regulation of senescence and suggest different therapeutic strategies to reduce mitochondrial oxidative stress. The authors conclude that a control of mitochondrial oxidative stress could greatly contribute to the effective control and treatment of aging and age-related diseases.
My biggest problem with this review is that it is not always clear how the authors use the term “senescence”. It is introduced as the loss of cells to proliferate (line 32-33), but then used to discuss age-related decline (chapter 3, beginning at line 126), and bioenergetic changes (line 218-220). A distinct definition of senescence is specifically important for chapter 4 (beginning in line 168) of this review. Please provide a definition of senescence and state what will be the focus of this review.
Line 94: The authors state that the generation of ROS leads to a depletion of Fis1. Please add a reference for this statement, because it is more commonly known that ablation of knockout of Fis1 leads to an increase of ROS generation.
Line 106/107: Reference is missing for this content: “Senescence is characterized by Ca2+ overloaded mitochondria and is exacerbated by mitochondrial Ca2+ overload-induced mitochondrial ROS generation”.
Line 154/155: The authors write “Furthermore, continuous influx through the mPTP causes electron leakage, which increases mitochondrial ROS production” but did not specify in the text of the figure what influx is meant.
This review uses frequently the word “excessive” without definition. What is excessive relative to not excessive?
Figure 1: It is not clear to me how Fis1 contributes to the appearance of large mitochondria?
Figures 2 and 3B should be combined. Fast opening and closings (flickering) of the mPTP have been discussed as a mechanism to release mitochondrial Ca2+. Therefore, the mPTP should be added into figure 2.
Figure 4: I find it confusing that CoQ and Cyt c are shown using the same shape and color. Is it possible to give Cyt c a different color?
Reviewer 2 Report
Yun Haeng Lee t al., review the potential impact of mitochondrial oxidative stress in senescence and the future pharmacological strategies for senescence treatment.
Some references and text editions may be incorporated. Also, I suggest the authors include additional data in Table 1. This table could contain ROS sources, the type of ROS described in these studies, and the enzymes involved in ROS generation. This information is valuable for the aim of this review in Antioxidant.
Mayor concerns
Section 2: Incorporate the reference: Mitochondria are required for pro-ageing features of the senescent phenotype EMBO J 2016 2016 Apr 1;35(7):724-42. This article describes several molecular signals associated with mitochondrial function and activated senescence.
Line 67: Modify this phrase: receptors including mitochondrial fission factor
(Mff) and mitochondrial fission 1 protein (FIS1) bind to dynamin-related protein 1 (Drp1)
The role of Fis1 in mitochondrial fission is still not completely clarified. Please, incorporate recent studies. Also, incorporate the reference: Tatjana Kleele et al., Distinct fission signatures predict mitochondrial degradation or biogenesis. 2021 May;593(7859):435-439
Include also MiD49, and MiD51 proteins in the mitochondrial fission mechanism description.
Line 79: Excessive levels of mitochondrial oxidative stress deplete the mito- 79
chondrial fission regulator FIS1, resulting in an imbalance in mitochondrial fusion/fission 80
[16].
Fis1 protein function in senescence was not included in this study. Incorporate reference:
Decreased expression of Drp1 and Fis1 mediates mitochondrial elongation in senescent cells and enhances resistance to oxidative stress through PINK1. Mai S, Klinkenberg M, Auburger G, Bereiter-Hahn J, Jendrach M. J Cell Sci. 2010 Mar 15;123(Pt 6):917-26
Line 139: Please check reference 26. This reference doesn’t describe the Fenton reaction or the link between hydroxyl radicals and mitochondria.
Line 143-155 In this paraph, the authors describe the role of calcium overload in mitochondrial dysfunction. The author should extend the current hypothesis about the calcium-dependent ROS generation into mitochondria. For instance, the authors could incorporate the effect of calcium on dehydrogenase activity and the byproduct superoxide.
Line 174: Reference 33 doesn’t demonstrate that ROS promotes PARP activation. Incorporate some additional references.
Line 215: Please cite the “original article” where was described the link between Aβ and mtDNA.
Line 241: Reference 55, this study was carried out in mice models.
Line 292: This phrase should be saying “Ataxia Telangiectasia Mutated (ATM)”.
Line 296: Check the phrase: Acidified lysosomes activate mitophagy. Reference 73 did not describe this concept. Modify the phrase or include an original article in which was described this specifically this mechanism.
Figure1. In this figure, the authors should incorporate MiD49 and MiD51 proteins.
Figure 2 Panel A and Panel B and Figure 3 Panel B. To make it easier to interpret these figures, please show the differential mechanism by which the protein described in the text modulates the mitochondrial activity in senescent cells, for example, increasing the size of the arrow. Only an increase in calcium is easy to visualize in figure 2.
Reviewer 3 Report
In this article, Lee et al discuss senescence-associated mitochondrial alterations and mitochondrial oxidative stress. In addition, they assess the importance of targeting mitochondrial oxidative stress for the regulation of senescence and suggest different therapeutic strategies to reduce mitochondrial oxidative stress. The conclude that the modulation of mitochondrial oxidative stress will contribute to the treatment of aging and age-related diseases.
The review is interesting and well written. Figures are also of high quality.
However, two main point should be addressed:
1) The disruption of metal homeostasis (iron) during senescence that may contribute to oxidative stress.
2) The contribution of mitochondrial dysfunction to inflammation and the proinflammatory secretory phenotype.
Reviewer 4 Report
In this review paper titled 'Targeting Mitochondrial Oxidative Stress as a Strategy to Treat Senescence', the author Lee et al discuss senescence-associated mitochondrial alterations and the resulting increase in mitochondrial oxidative stress. The authors did an excellent job summarizing the literature and presenting the main conclusion. Some revisions are recommended. See detailed comments below.
1. Fig. 1, the authors should point out the localization of the proteins involved in fusion and fission on the mitochondrial membranes. Similar to those of Fig. 2.
2. Some of the papers cited are not original studies. The authors should try citing the original studies or at least well recognized review papers. Example, ref. 26, 46, 51
3. The discussion of ROS is not sufficient. ROS can be deleterious but are also integral for cell survival and adaptive signaling.
4. There have been some recent studies that show that several organisms have extended lifespans when mitochondrial ROS production is high for short periods of time (Scialo et al 2016), contrary to the main conclusion in this review. The authors might want to acknowledge that and discuss the potential implications.
5. line 113, citations are needed.
6. line 153-155, citations are needed.
7. ref. 52, the cited study appears to talk about the maximum life span in the heart and brain of mammals.
8. The relationship between mitochondrial oxidative stress and senescence is a feedback loop, which should be reflected by the title of section 3.
9. line 198, the authors should be more specific about the proteins involved in proteostasis.
10. Fig.4B needs improvement. The authors should clearly label the components of mitophagy machinery in the figure. The ubiquitin appears to be unproportionally big in the figure.
11. The authors' focus on PTEN-induced PINK1-Parkin-dependent mitophagy pathway. How about other mitophagy pathways?
12. The author should make a summery figure to summarize the relationship between mitochondrial ROS and cell senescenece (mtDNA damage, proteostasis change etc).
13. The authors should also briefly discuss the future perspective and challenges in the field in the final section.
Round 2
Reviewer 1 Report
The authors have improved the manuscript significantly.
Author Response
We appreciate the opportunity to revise our manuscript (Antioxidants-2296330) entitled “Targeting Mitochondrial Oxidative Stress as a Strategy to Treat Aging and Age-related diseases”. The Reviewer provided insightful comments to improve this manuscript. Accordingly, we modified our manuscript by incorporating his/her valuable suggestions. All text has been spell-checked and corrected.
Reviewer 2 Report
typo
Line 55: ang
Line 260: proteostasis
Table 3: brain mito-chondria
from male Wistar rats
Table 2: cluster biosynthe-sis
Reviewer 3 Report
The authors have addressed all my concerns
Author Response

(The authors gave the same response as above.)

Reviewer 4 Report
The authors did an excellent job revising the manuscript. All my concerns and comments are adequately addressed. I recommend accept in present form.
Author Response

(The authors gave the same response as above.)
